# Integrated energy storage and $CO_2$ conversion using an aqueous battery with tamed asymmetric reactions

Yumei Liu[1], Yun An[1], Jiexin Zhu [1,2], Lujun Zhu[1], Xiaomei Li[3], Peng Gao[3], Guanjie He [2] & Quanquan Pang [1] ✉

Developing a $CO_2$-utilization and energy-storage integrated system possesses great advantages for carbon- and energy-intensive industries. Efforts have been made to developing the Zn-$CO_2$ batteries, but access to long cycling life and low charging voltage remains a grand challenge. Here we unambiguously show such inefficiencies originate from the high-barrier oxygen evolution reaction on charge, and by recharging the battery via oxidation of reducing molecules, Faradaic efficiency-enhanced $CO_2$ reduction and low-overpotential battery regeneration can be simultaneously achieved. Showcased by using hydrazine oxidation, our battery demonstrates a long life over 1000 hours with a charging voltage as low as 1.2 V. The low charging voltage and formation of gaseous product upon hydrazine oxidation are the key to stabilize the catalyst over cycling. Our findings suggest that by fundamentally taming the asymmetric reactions, aqueous batteries are viable tools to achieve integrated energy storage and $CO_2$ conversion that is economical, highly energy efficient, and scalable.

The conventional energy-intensive industrial sectors, particularly the power plants, steel- and cement-making as well as fertilizer industries, face pressure to reduce carbon emission and embrace renewable energies simultaneously[1–3]. The former calls for effective on-site carbon dioxide ($CO_2$) capture or upgrading to valuable chemicals[3–5], and the latter relies on flexible energy storage to level the power load[2,6]. There has been tremendous progress on electrocatalytic $CO_2$ reduction to valuable chemicals (CO, formic acid, ethylene etc.)[3,7,8] and that on-site implementation of lithium-ion battery packs is close to being commercially available[9]. However, having two separate implements would incur large capital and space investments[2,10,11], especially in view of the energy input involved in breaking the thermodynamically very stable C=O bonds in $CO_2$ (806 kJ $mol^{-1}$)[12]. It therefore stands great significance if one can integrate the two seemingly irrelevant systems and economically fulfill value-added utilization of $CO_2$ and storage of renewable energies.

An aqueous battery involving $CO_2$ reduction reaction (CRR) holds such promise. For realizing such an overarching goal, the battery should involve asymmetric reactions on discharge and charge, meaning $CO_2$ is reduced to a valuable chemical (with appreciable voltage) in one direction and another chemical (not the reduction products) is oxidized in the other. By doing so, the device affords synergistic $CO_2$ conversion and energy storage. Apparently, an electrochemical cell with switchable electrolyzer/fuel modes based on interconversion between $CO_2$ and formate can only function for energy storage[13], as is the case in the conventional organic lithium/sodium-$CO_2$ batteries systems based on the reversible conversion between $CO_2$ and oxalates[14,15]. Also, aqueous media promises greener, safer, and lower-cost devices than the organic solvents[16]. Zinc is an earth-abundant metal and can be plated/stripped in aqueous solutions unlike the highly active alkali metals (Li, Na, and K)[16,17]. As such, aqueous zinc batteries that exploits $CO_2$ reduction upon discharge (the so-called

[1]Beijing Key Laboratory for Theory and Technology of Advanced Battery Materials, School of Materials Science and Engineering, Peking University, 100871 Beijing, China. [2]Christopher Ingold Laboratory, Department of Chemistry, University College London, London WC1H 0AJ, UK. [3]International Centre for Quantum Materials, Collaborative Innovation Centre of Quantum Matter, Peking University, Beijing, China. ✉e-mail: qqpang@pku.edu.cn

Zn-CO$_2$ battery) could achieve integrated CO$_2$ conversion and energy storage[16], if recharging of the battery (i.e. regeneration of the anode) occurs economically through designed oxidation reactions (schematically shown in Fig. 1a). Further, such oxidation can be designed to utilize oxidizable species that are ubiquitous in toxic industrial wastewater from thermal and nuclear power plants as well as other industrial sectors (e.g. hydrazine hydrate)[18–20]. Therefore, this innovative technology also stands out with obvious economic benefits for wastewater treatment compared to the current Zn-CO$_2$ batteries, and highlights great contribution to the cross-field of carbon-energy-environment nexus.

The oxidation reaction on recharging of the Zn-CO$_2$ battery has almost been exclusively the oxygen evolution reaction (OER) despite CRR has been versatile[21–25]. Up to now, almost all efforts have been dedicated to developing bifunctional catalysts to promote and stabilize CRR and OER. However, these efforts have been fruitless as the sluggish OER on recharging, by nature, causes a highly oxidative environment undesired for stable operation and also requires a high oxidation potential (i.e. high overpotential). For example, the batteries using defected single-atom Fe-N-C, bimetallic Ni-Fe or Sn-metallene catalysts exhibited an overly high charging voltage of 2.23-2.7 V[22,24,26]. Recent efforts using Fe-N-C or Cu-Ni bimetallic or SnO$_2$/MXene catalysts have enabled low charging voltages of 1.2-1.7 V at 0.5-2 mA cm$^{-2}$, but for an operation duration of only 25-60 h[23,25,27], presumably due to the occurrence of undesired side reactions during charging (e.g. oxidation of unstable substances on the catalysts). These results demonstrate a formidable activity–stability trade-off for such delicate systems. In fact, given the high overpotential incurred by OER[28], altering of CRR and OER upon dis(charge) significantly disturbs the redox environment around the catalyst, which can cause a permanent

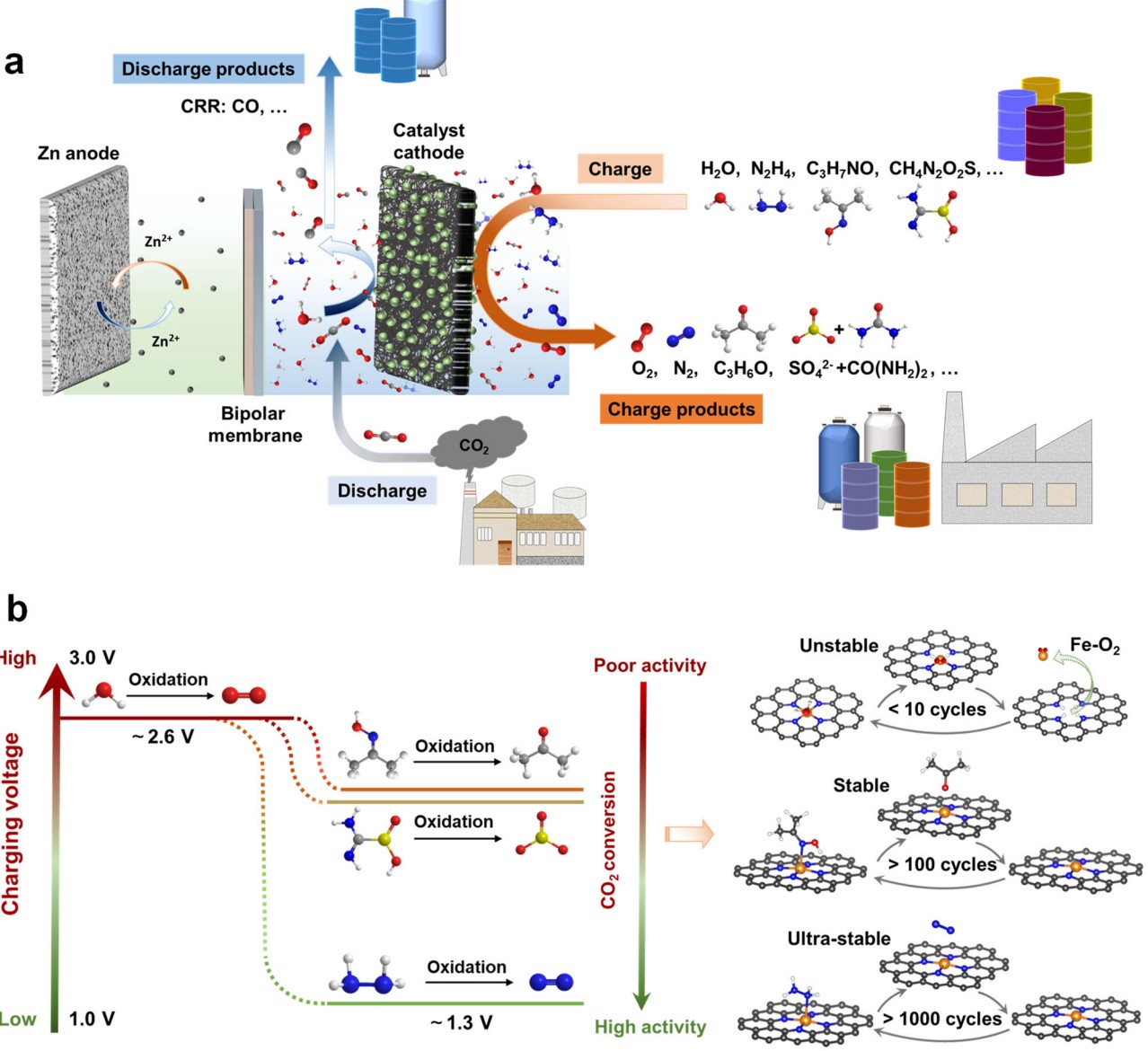

**Fig. 1 | Concept of the Zn-CRR/RMOR battery device. a** Illustration of the batteries with CRR on discharge, and the respective OER or RMORs on charge. **b** Schematics showing the concerted effect of lower charging voltages and promoted catalyst stability with the implementation of RMORs, as well as the underlying mechanism from the view of energy landscape. In contrast to the case of OER with a life of <10 cycles (denoted as unstable), the oxidation of ACTO leads to the enhanced stability with a life of >100 cycles (denoted as stable), while HzOR leads to a super long-term stability with a life of >1000 cycles (denoted as ultra-stable) (**b**). The schematic is meant to show the main oxidation products of the reducing molecules. The red, white, gray, blue, orange, and yellow balls represent O, H, C, N, Fe, and S atoms, respectively.

structural degradation to the catalyst, leading to a grand challenge on the cycle life.

Pivoting from the wisdom of tailoring the catalysts, herein, we propose to fundamentally alter the recharging oxidation reaction to unleash the harsh requirement on the catalyst and eliminate its rapid degradation. This is the rationale behind the short cycling life in the reported Zn-$CO_2$ batteries, as we now for the first time discover and discuss in our work. We solve this challenge by replacing the OER with reducing molecules oxidation reactions (RMORs) which also significantly lowers the charging voltage. We show that the strategy is universal for three representative reducing molecules, namely hydrazine (Hz), acetone oxime (ACTO), and thiourea dioxide (TUDO) (Fig. 1a). In particular, by coupling hydrazine oxidation (HzOR) with CRR, the Zn-CRR/HzOR battery shows stable operation for an unprecedented long period of over 1000 h at 1.43 mA cm$^{-2}$ and charging voltage as low as 1.2 V. Notably, the demonstrated battery lifespan using HzOR increases by 50-fold compared to that of using OER, showing 90% Faradaic efficiency for CO ($FE_{CO}$) at 418 h. As proposed by ab initio simulations and spectroscopic analyses, we discover that Fe demetallation from the Fe-N-C catalyst is the inherent cause of catalyst degradation in conventional OER-based Zn-$CO_2$ battery, while our Zn-CRR/RMORs batteries can well stabilize the active $FeN_4$ sites. Meanwhile, the reducing molecules weaken the *CO-Fe interaction, promote the desorption of *CO during CRR, and inhibit the competing hydrogen generation, thereby contributing to slightly improved CRR activity and selectivity. Critically, the oxidized product is gaseous $N_2$ that can be expelled in real time, thus enabling flow cell design and promising continuous, scalable, and modularized service. Our work pioneers in fulfilling the functions of energy storage, $CO_2$ conversion, and wastewater treatment in an aqueous battery, making a significant contribution to the technological revolution through interdisciplinary efforts.

## Results

### Concept of coupling RMORs with CRR

Our aqueous Zn-CRR/RMOR battery uses a Fe-N-C catalyst cathode and a Zn plate anode, separated by a bipolar membrane. A 0.5 M $CO_2$-saturated $KHCO_3$ solution and a 1 M KOH + 0.2 M $Zn(CH_3COO)_2$ solution were employed as catholyte and anolyte, respectively. As illustrated in Fig. 1a, during discharge, the conversion of $CO_2$ (e.g. from pretreated flue gas) to CO occurs in the cathode compartment while the Zn plate is dissolved into the anolyte, affording a voltage of 0.5 V at 0.71 mA cm$^{-2}$. During charge, the reducing molecules are oxidized to generate $N_2$ gas, acetone or sulfate, and the $Zn^{2+}$ is plated back to the Zn anode, completing replenishment of the battery.

In contrast to the conventionally configured Zn-$CO_2$ battery that relies on high-potential and kinetically sluggish OER on charge, by using tamed RMOR, our Zn-CRR/RMOR battery maintains the chemical integrity of the $FeN_4$ active sites, therefore exhibiting outstanding stability with a low charging voltage (Fig. 1b). The asymmetric redox reactions enable synergistic $CO_2$ conversion, energy storage, and wastewater treatment that can be coordinated on a daily basis in practice, here taking a cement-making factory for example. In the day, when carbon emission and power demands are intensive, the Zn-CRR/RMOR system undergoes discharge thus outputting energy and converting $CO_2$ to chemicals, while in the night, the system recharges to oxidize hydrazine and to replenish the zinc anode. According to our preliminary techno-economic assessment, this technology shows obvious economic advantages compared to that of the classical $LiFePO_4$ battery, the Zn-$CO_2$ battery, and the electrochemical CRR system (see Supplementary Fig. 1 and Note 1 for more details).

To achieve this goal, we identified and optimized an atomically dispersed Fe-N-C catalyst for catalyzing the reduction of $CO_2$ to CO, and the oxidation of reducing molecules. The detailed synthesis and physical characterizations are discussed in Supplementary Figs. 2–4,

Table 1, and Note 2. Gaseous and liquid products were quantitatively analyzed by using online gas chromatography (GC) and [1]H nuclear magnetic resonance (NMR), respectively. For the conventional CRR device (three-electrode cell), the result of product analysis demonstrates that only gaseous products ($H_2$ and CO) can be detected without any liquid product (Supplementary Fig. 5). The obtained Fe-N-C catalyst demonstrates remarkable selectivity for CO generation at low over-potentials with $FE_{CO}$ over 90% from −0.3 to −0.6 V vs. the reversible hydrogen electrode (RHE, used throughout the article), with peak $FE_{CO}$ of 98% at −0.4 V vs. RHE (Supplementary Fig. 5b). For constructing the aqueous Zn-CRR/RMOR batteries, we note that increasing the catalyst loading mass shows little correlation with $FE_{CO}$, albeit leading to a slight increase in voltages (Supplementary Fig. 6 and Note 3). For a comprehensive evaluation, the loading mass is established to be 1 mg cm$^{-2}$. The assembled zinc battery shows a discharging voltage of 0.5 V and a $FE_{CO}$ as high as 98% at current densities ranging from 0.71 to 12.86 mA cm$^{-2}$, confirming high CRR performances.

### Fundamental catalytic behavior with the reducing molecules

We first examined the impact of the reducing molecules on the Fe-N-C catalyzed CRR and the oxidation reactions in three-electrode cells using the linear sweep voltammetry (LSV). The presence of reducing molecules shows slight impact on the CRR current response with saturated $CO_2$ (Fig. 2a). For oxidation, we denote the oxidation of Hz, ACTO, and TUDO as HzOR, ACTOR, and TUDOR, respectively. For a current density of 10 mA cm$^{-2}$, HzOR shows an oxidation potential of only 0.38 V, and ACTOR and TUDOR show 1.18 V and 1.25 V, respectively, all of which are lower than 1.57 V for OER (Fig. 2b). Note that the difference in activity of these oxidation reactions can be explained by the charge transfer capability as assessed by the electrochemical impedance spectroscopy (EIS)[29,30]. The charge-transfer resistance ($R_{ct}$) of HzOR is only about 1/3, 1/4, and 1/8 of that in ACTOR, TUDOR, and OER (Fig. 2c). An increase in the concentration of reducing molecules can lower the oxidation potential and increase the current response, particularly for HzOR (Supplementary Fig. 7).

For constructing the Zn-CRR/RMOR batteries, the impact of reducing molecule concentrations on the voltages and $FE_{CO}$ was also investigated (Fig. 2d, e), wherein the gaseous products were periodically quantitated via online GC. While the $FE_{CO}$ decreases from 77% to 54% as TUDO concentration increases from 2.6 to 10.4 mM, the voltages and $FE_{CO}$ can be well-maintained within a wide range of concentrations of ACTO (13–334 mM) and Hz (23–344 mM). Notably, high $FE_{CO}$ of 97% and very stable voltages can be retained for HzOR even when the Hz concentration increases by 15-folds (Fig. 2d, e). Interestingly, as Hz concentration rises to 459 and 688 mM, the charging voltage can be further lowered but with a decrease in $FE_{CO}$ to 80% and a decrease in output voltages to 0.25 V (Supplementary Fig. 8).

### Performance of the Zn-CRR/RMOR batteries

Our Fe-N-C catalyst exhibits excellent long-term durability in a unidirectional CRR cell (up to 237 h with $FE_{CO}$ over 95%) or OER cell (Supplementary Fig. 9). However, a full Zn-$CO_2$ battery involving OER shows a very short cycling life and rapid decline in $FE_{CO}$ (Fig. 3a, b). In great contrast, by replacing OER with RMORs, the Zn-CRR/RMOR battery shows superior rechargeability and high $FE_{CO}$ of 92% over 1000 cycles, as discussed below (Fig. 3a, b).

At 4.29 mA cm$^{-2}$, HzOR shows a much lower charging voltage (1.35 V) than OER (2.45 V), and so are TUDOR (2.0 V) as well as ACTOR (2.1 V) (Fig. 3a). The extremely stable voltage evolution for the batteries using RMORs is illustrated in Fig. 3a, c and Supplementary Fig. 10. Further, with OER on recharging, the $FE_{CO}$ quickly decreased from 98% to 80% and retained only 14% after 22 cycles (Fig. 3b). In sharp contrast, the periodically measured $FE_{CO}$ for the Zn-CRR/HzOR battery maintained about 97% and 92% $FE_{CO}$ at 100 and 1000 cycles (Fig. 3b). Note that the cycling life obtained for the Zn-CRR/RMOR batteries follows

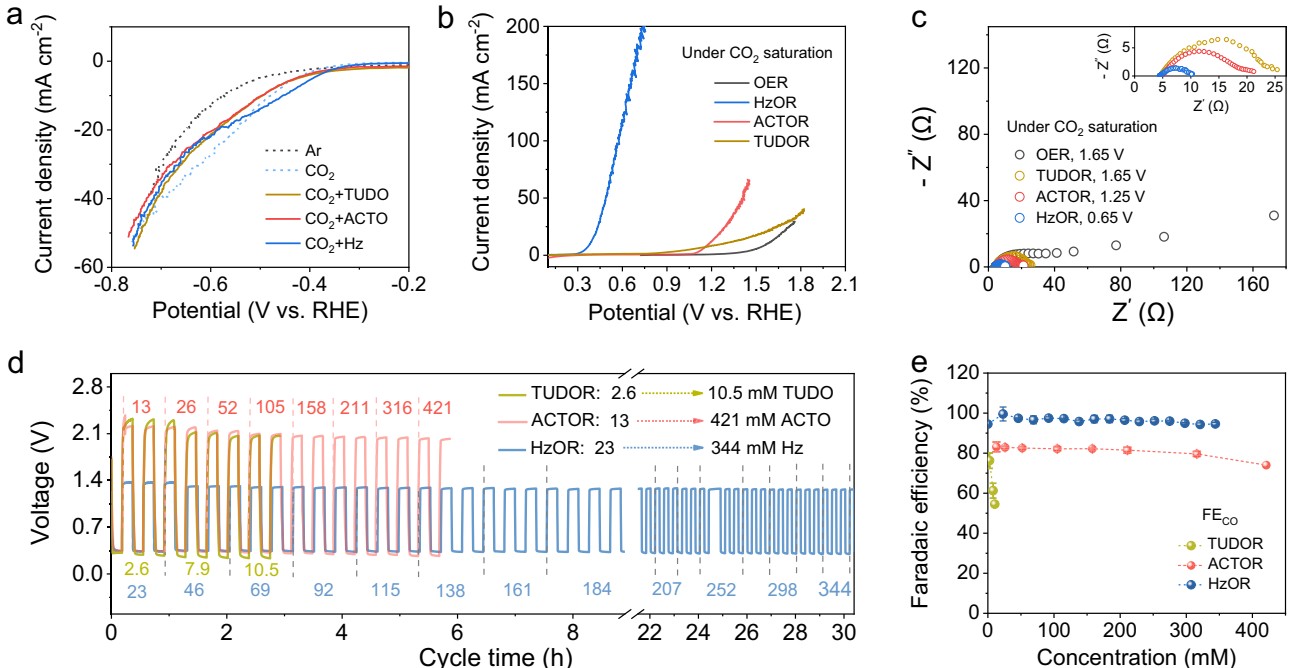

**Fig. 2 | The fundamental catalytic behaviors by coupling RMORs with CRR.** **a** LSV curves of Fe-N-C catalyzed CRR without and with the presence of reducing molecules (Ar saturated solution is included for reference and $CO_2$ is saturated in all other cases). **b** LSV curves of Fe-N-C catalyzed oxidation reactions with different reducing molecules *vs.* OER. **c** EIS plots obtained with different oxidation processes at respectively noted potentials. Evaluation of the Zn-CRR/RMOR batteries voltages (**d**) and $FE_{CO}$ (**e**) in response to varied concentrations of the reducing molecules. The numbers noted in **d** represent concentrations of TUDO (in yellow), ACTO (in red), and Hz (in blue); the error bars in **e** are the standard deviations from multiple measurements.

the order of HzOR (1140 cycles, 418.0 h) >ACTOR (369 cycles, 135.3 h) >TUDOR (180 cycles, 65.8 h). In addition, the presence of $Zn^{2+}$ in catholyte has little effect on the electrochemical behavior of our Zn-CRR/HzOR battery, thus eliminating the concern on possible $Zn^{2+}$ crossover during cycles, as discussed in Supplementary Fig. 11 and Note 4. Further, the Zn-CRR/HzOR battery presents superior rate capability along with an impressive selectivity toward CO generation (Fig. 3d). High $FE_{CO}$ of ~96% in a wide range of current densities from 2.86 to 14.29 mA cm$^{-2}$ is achieved. Also, the charging voltage rises from only 1.15 to 1.70 V when current density increases by 20-folds, demonstrating excellent reaction kinetics for CRR and HzOR (Fig. 3d). Figure 3e shows that a desirable $FE_{CO}$ of 90% over 850 cycles can be achieved at a high current density of 7.14 mA cm$^{-2}$. Notably, the battery shows an outstanding cycling life of 2768 cycles (1015.8 h) at a moderate current density of 1.43 mA cm$^{-2}$ (Fig. 3f), with high $FE_{CO}$ retention (90% of the initial $FE_{CO}$) and steady voltages (charge: 1.35 V; discharge: 0.38 V; Supplementary Fig. 12). These matrixes are far beyond the reach of any reported Zn-$CO_2$ batteries[21–27,29–33], with more details summarized in Supplementary Table 2. All the discharge and charge products are determined using online GC and $^1$H NMR measurements, as summarized in Supplementary Table 3, Figs. 13–15, and Notes 5, 6. In addition, the Faradaic efficiency of $H_2$ over cycling at various current densities are also provided in Supplementary Fig. 16.

Such excellent electrochemical performance can be attributed to the low potential of RMOR that prevents the catalyst from degradating at high potential and ensure structural stability and efficient CRR. Following the same strategy, we also demonstrate the success of promoting another type of CRR by coupling with RMORs, as showcased by the $CO_2$ reduction to formic acid with a much higher FE using a Bi-based catalyst (Supplementary Fig. 17). This confirms the generalizability of our configuration to a variety of CRRs where RMORs suppresses degradation of the catalysts. It is known that zinc anode is plagued by the dendrite growth and hydrogen evolution (Supplementary Fig. 18). Herein instead of large dendrites, we observed

the formation of mossy fibrous deposits with a small size in the Zn‖Zn symmetric battery (Supplementary Fig. 19c, d). To further improve the zinc anode and fundamentally alter its plating behavior, we formulated a functionalized electrolyte (3 M KOH + 1.4 M KF + 0.75 M $K_2CO_3$ + 0.032 M ZnO, denoted as 3 M KOH + KFCZnO) that supports a much more uniform and flatter Zn stripping/platting in the Zn‖Zn symmetric battery and Zn-CRR/HzOR battery (Supplementary Figs. 19e, f, and 21d–f). Due to the suppressed hydrogen evolution, less dead zinc, and lower concentration polarization, enhanced rate capabilities with stable cycling performance were obtained. In particular, the discharged voltage was promoted from 0.32 V to 0.44 V at 4.29 mA cm$^{-2}$ (see Supplementary Figs. 18–21 and Notes 7–10 for more details).

## Mechanistic insights on enhanced CRR activity

To reveal the mechanism of reducing molecules effect on CRR, density functional theory (DFT) calculations were performed to visualize the CRR process on $FeN_4$ activity centers (the $FeN_4$ configuration is confirmed below). The negative adsorption energy of ACTO (−0.72 eV) and Hz (−0.74 eV) indicates that their adsorption on the catalyst surface is energetically favorable. We first show that the less distorted $FeN_4$ sites upon adsorbing *COOH and *CO with the presence of Hz or ACTO account for the high CRR selectivity (discussed in Fig. 4a, Supplementary Fig. 22, and Note 11). Following the adsorption and activation of $CO_2$ molecules, the blank $FeN_4$ sites apparently show a larger distortion, which results in a stronger interaction between Fe and *CO, making it difficult to release CO. In contrast, with the presence of Hz or ACTO, there is less distortion of the $FeN_4$ site indicating weaker interaction with the *CO that drives its facile desorption. This is owed to the electron-donating feature of the reducing molecules added, which means that the strategy employed here synergistically benefits the low-potential oxidation and high-activity CRR. This is critical for the success of our concept of coupling RMOR with CRR in a single bidirectional battery.

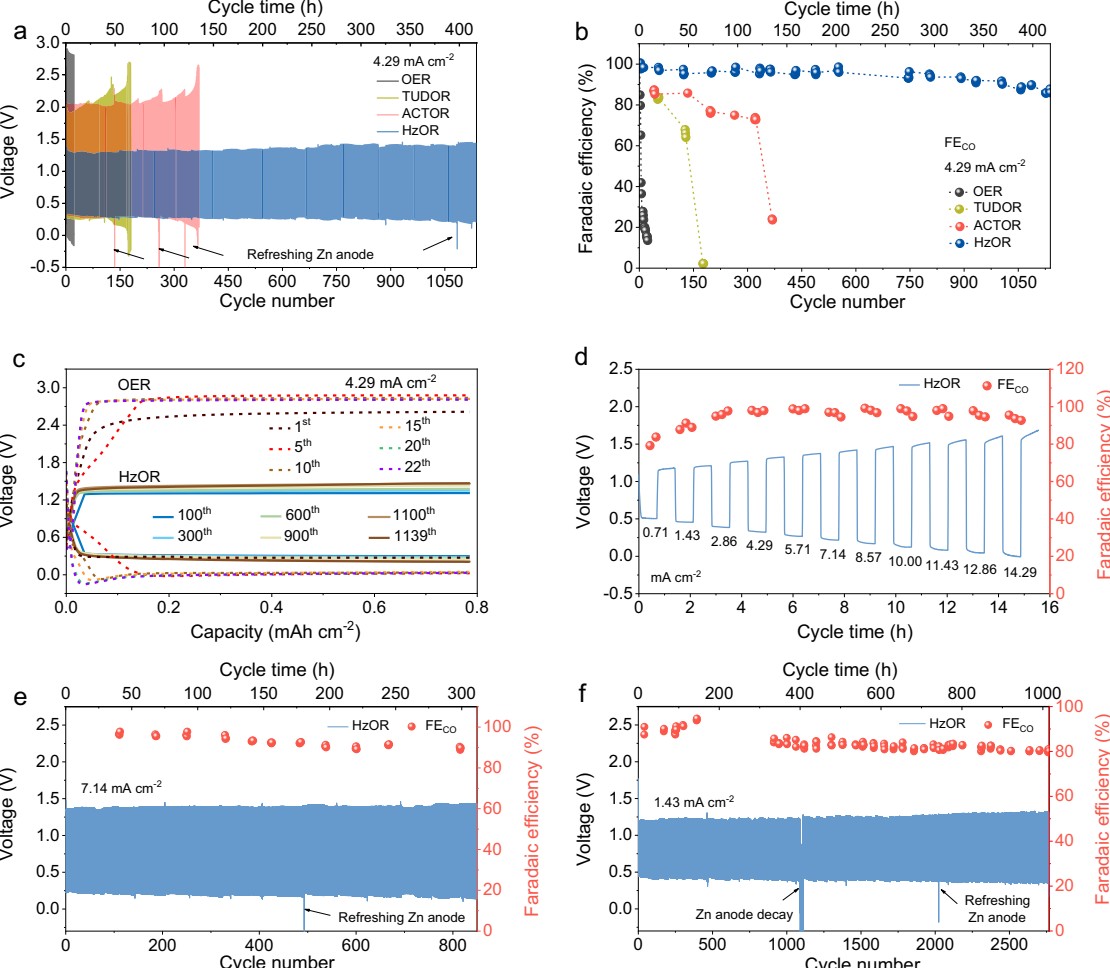

**Fig. 3 | Electrochemical performances of the Zn-CRR/RMOR batteries.** The battery voltage evolution (**a**) and the corresponding $FE_{CO}$ (**b**) of the Zn-CRR/RMOR batteries with varied RMORs at 4.29 mA cm$^{-2}$ (with OER as the reference). The initial concentrations of TUDO, ACTO, and Hz in **a** and **b** are 7.9, 79, and 92 mM, respectively, and their daily addition amount are 5.2, 26, and 46 mM.
**c** Galvanostatic discharge/charge voltage profiles showing a much lower voltage polarization in Zn-CRR/HzOR batteries than in the Zn-CO$_2$ battery involving OER.
**d** The rate capability measurements showing the voltages of the Zn-CRR/HzOR battery with varied current densities (69 mM Hz). Long-term cycling performance

of the Zn-CRR/HzOR battery demonstrating continued high selectivity for CO and voltage stability at 7.14 mA cm$^{-2}$ (**e**) and 1.43 mA cm$^{-2}$ (**f**). The initial concentration of Hz in **e** and **f** is 92 mM, and the daily addition amounts are 23 and 46 mM for **e** and **f**, respectively. The $FE_{CO}$ was periodically measured for all systems to ensure a proper understanding of the temporal evolution of CRR; the arrows indicate when the Zn anode and anolyte are refreshed, which in the case of HzOR occurs every ten, seven, five days for measurements at 1.43, 4.29, and 7.14 mA cm$^{-2}$, respectively. For all batteries, each cycle includes 11 min charge and 11 min discharge.

The finding is further supported by the differential charge density analyses of the *CO adsorbed on the FeN$_4$ sites wherein the charge transfer (between CO and FeN$_4$) is less in Hz-FeN$_4$ (0.15 e$^-$) and ACTO-FeN$_4$ (0.13 e$^-$) than in blank FeN$_4$ (0.27 e$^-$) (Fig. 4b). A less charge density transfer means a weaker binding strength for CO and hence a lower energy barrier for its desorption[22]. We note that ACTO loses it binding to the Fe atom once COOH* and CO* are formed, whereas Hz maintains the close binding with Fe throughout the CRR process, thereby exhibiting the least distortion of FeN$_4$ sites among all systems studied.

Consistent with previous reports, we also found that CRR on the FeN$_4$ sites is governed by the desorption of *CO due to the strong binding between *CO and Fe (Fig. 4c)[25,26,31]. With the presence of Hz or ACTO, *CO desorption shows a slightly lower energy barrier than that in the blank FeN$_4$ (1.02 eV vs. 1.10 eV), indicating the reducing molecules do not incur further energy penalty and may slightly promote the intrinsic activity of CO$_2$-to-CO reduction. Furthermore, HER as a dominant competing reaction upon CRR, was investigated. A higher energy barrier for the formation of H* is observed on the FeN$_4$ site with the presence of Hz and ACTO (Fig. 4d), indicating that Hz and ACTO

have an inhibiting effect on H$_2$ generation. We further quantified the limiting potential difference between CRR and HER, i.e. the $U_L(CO_2)-U_L(H_2)$, an indicator for CRR selectivity (Fig. 4e)[21,26]. Compared to the blank FeN$_4$ (−0.88 eV), more positive values for the Hz-FeN$_4$ (−0.7 eV) and ACTO-FeN$_4$ (−0.6 eV) sites suggest the higher selectivity for CO formation[21,26]. As for TUDO, our proton-releasing experiments suggest that the H$^+$ released from TUDO leads to a lower $FE_{CO}$ compared to that on HzOR-FeN$_4$ and ACTOR-FeN$_4$ sites (Supplementary Fig. 23 and Note 12).

## Mechanistic insights on the long-term stability
We then discuss the mechanism of the significantly promoted long-term stability of the batteries. First, in great contrast to the almost identical voltage profiles (first four cycles) for the Zn-CRR/RMOR batteries, the one with OER shows a sudden increase in charging voltage on the 3$^{rd}$ cycle (2.4–2.8 V) (Fig. 5a, Supplementary Fig. 24). This is clearly shown by an additional oxidation peak in the d$Q$/d$V$ curve (Fig. 5b). We attribute such phenomenon to the degradation of catalyst, that is, electrochemical Fe demetallation occurring during OER, which has been found in ORR/OER systems[34–37]. Specifically, the

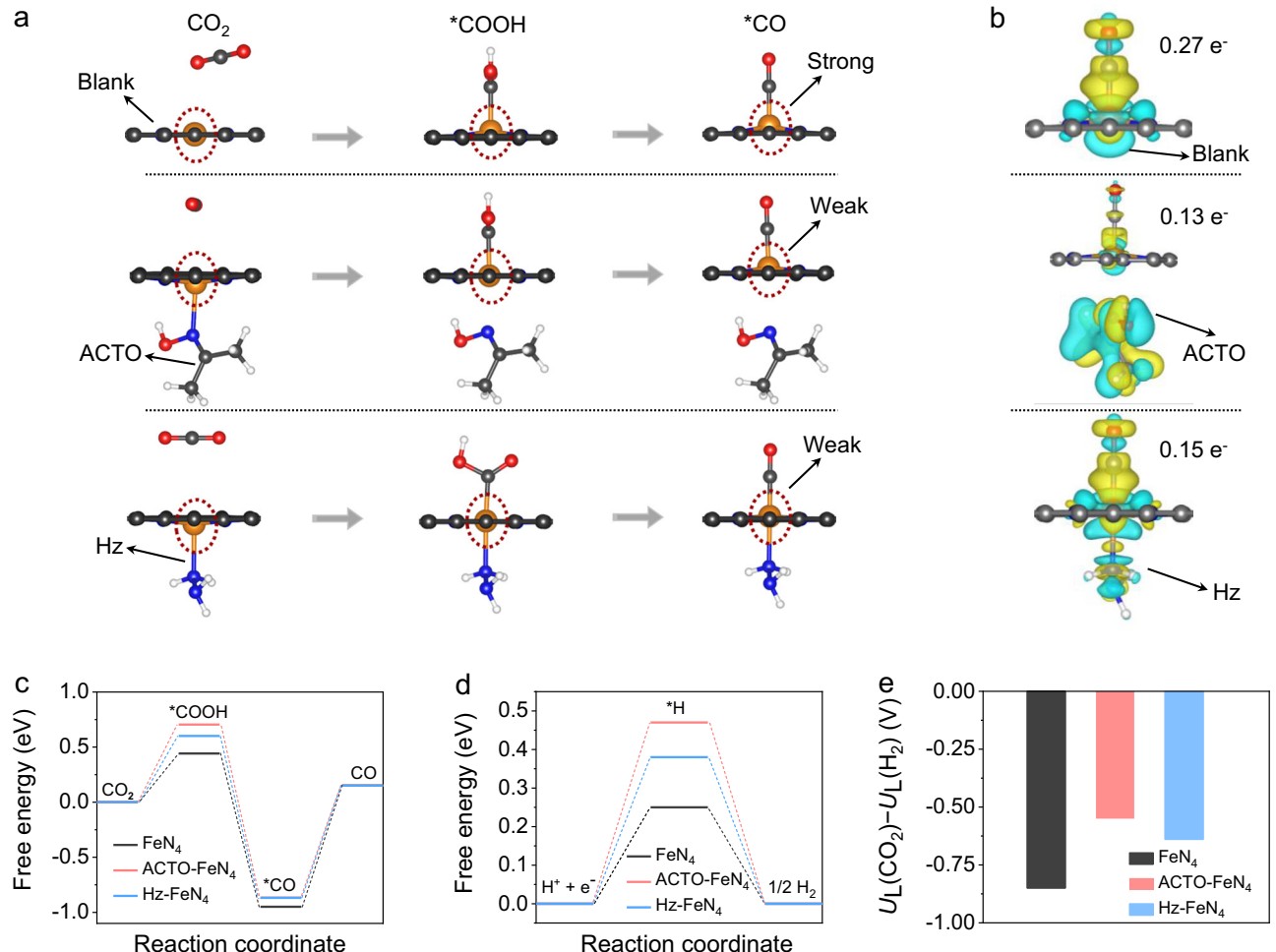

**Fig. 4 | DFT simulations of the $CO_2$-to-CO reaction on the Fe-N-C catalyst. a** The calculated $FeN_4$ site configurations with reducing molecules adsorbed during the CRR process (orange, blue, gray, red, and pink spheres stand for Fe, N, C, O, and H atoms). **b** The calculated charge density distribution of CO* adsorbed on $FeN_4$ sites (yellow and cyan areas represent accumulation and depletion of charge). **c**–**e** Calculated free energy diagrams for the CRR (**c**) and HER (**d**) processes on the $FeN_4$ sites with or without the reducing molecules, and the corresponding $U_L(CO_2) - U_L(H_2)$ values (**e**), an indicator for CRR selectivity.

reactive oxygen species OH* produced during OER first adsorbs on $FeN_4$ sites, and the OH* weakens Fe−N bonding and triggers the leaching of Fe[34,35]. To verify the stability of $FeN_4$ sites against demetallation, we assembled batteries that recharge by alternating OER and HzOR (Supplementary Fig. 25). In all cases when OER was introduced, a sharp decline in $FE_{CO}$ occurred; and critically, even when HzOR was introduced back on charge, the discharging voltage and $FE_{CO}$ remained extremely low, suggesting permanent damage to the catalyst by OER. We thus unambiguously show this leads to its irreversible sharp decline in the CRR activity over cycling of a $Zn-CO_2$ battery.

We performed Fe K-edge X-ray absorption spectroscopy (XAS) studies to examine the local chemical and structural evolution of the Fe-N-C catalyst over long-term cycling. The X-ray absorption near-edge structure (XANES) spectrum for the initial cathode locates at energies between those of $Fe_2O_3$ and $Fe_3O_4$, indicating the average valence of Fe in Fe-N-C is between +2 and +3. Locating at a more positive energy indicates a more oxidized state for the Fe[37,38]. After 100 cycles, the near-edge of the OER-based cathode locates at a higher energy than HzOR- and ACTOR-based cathodes, suggesting higher oxidation state of Fe (Fig. 5c, Supplementary Fig. 26 and Note 13). We further observe a trend of shifting to lower energy along cycling for the HzOR-based cathode, showcased by the spectra after 50 and 100 cycles (Fig. 5d). This indicates that Fe is further immune to being oxidized over cycling, which can be ascribed to the reducing environment introduced by CRR[38]. The evolution of Fe valence is related to the structural changes,

as further discussed below. Moreover, the K-edge spectra for HzOR-based cathodes at charged states almost overlap with those at discharged states, suggesting excellent catalyst stability that is immune to both reduction and oxidation (Fig. 5d).

The extended X-ray absorption fine structure (EXAFS) fittings further reveal the local structural configuration of Fe in the catalyst. The k- and R-space spectra of the initial catalyst are well-fitted with atomic dispersion of Fe coordinated with four N atoms in the first shell, indicating a predominant $FeN_4$ mode in Fe-N-C (Supplementary Fig. 27 and Table 4)[21,26]. Over cycling, the OER-based cathode shows a greater Fe-N bonding length (1.48 Å) whereas the ACTOR- and HzOR-based cathodes show similar and even shorter Fe-N bonding lengths than the pristine (1.44 Å, 1.42 Å; Fig. 5e, f, fitted spectra in Supplementary Fig. 27). This is further confirmed by the wavelet transform (WT)-EXAFS analyses (Fig. 5g–i, Supplementary Fig. 28). Clearly, the k-space intensity maximum for the ACTOR-cathode ($4.6 Å^{-1}$) and HzOR-cathode ($4.4 Å^{-1}$) is close to that for the initial cathode ($4.6 Å^{-1}$), in great contrast to the one for OER-based cathode ($5.6 Å^{-1}$; Fig. 5g–i, Supplementary Fig. 28). Such distinction could be related to the subtle differences in the local coordination environments of Fe atoms in these cathodes[28,36,39,40], as further discussed in Supplementary Note 13. In fact, the Fe-N-C catalyst contains two types of $FeN_4$ sites, the pyridinic N-coordinated site (denoted as S1) and the pyrrolic N-coordinated site (denoted as S2)[34,38,41]. For the OER-based cathode, the interaction of Fe (of S1) with *OH radicals causes its Fe

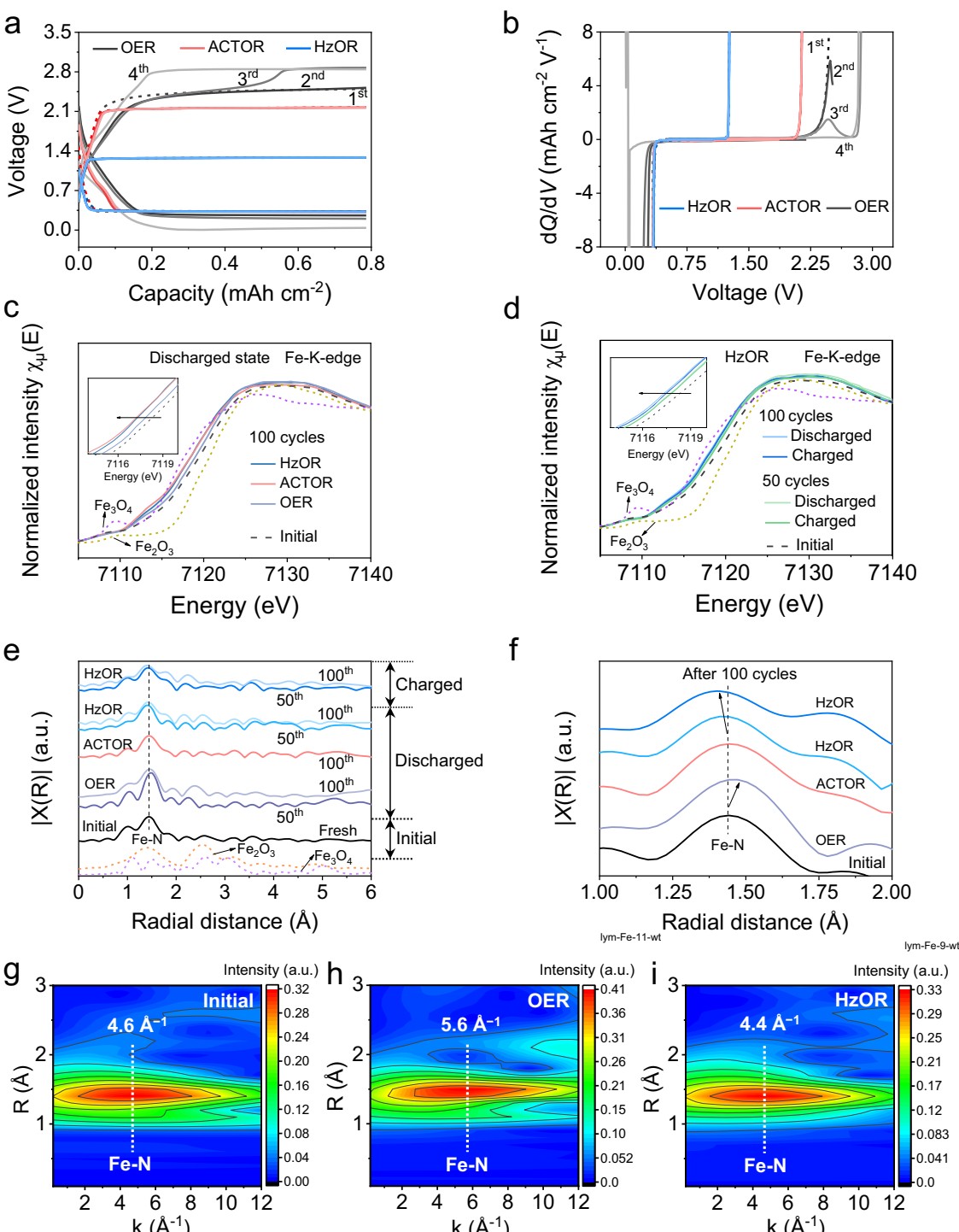

**Fig. 5 | Analyses of structural stability over long-term cycling.** Initial four galvanostatic discharge/charge profiles (**a**) showing a sudden increase in charging voltage for the OER-based battery, and their corresponding d*Q*/d*V* profiles (**b**) during charging. **c** Fe K-edge XANES spectra of Fe-N-C indicating a more positive valence of Fe in OER-based cathode at discharged state. **d** Fe K-edge XANES spectra of HzOR-based cathodes at charged and discharged states. The insets of (**c**) and (**d**) are the enlarged Fe K-edge XANES spectra. **e, f** Fourier transformation (FT)-EXAFS spectra of cathodes after cycling, displaying Fe-N coordination with magnification in **f**. **g–i** Fe wavelet transform (WT)-EXAFS spectra of Fe for the initial, OER- and HzOR-based cathodes after 50 cycles; the color bar represents the intensity.

demetallation upon cycling (Fig. 5a, b)[34,35], leaving majorly S2-Fe. Such coordination possesses a higher Fe oxidation state[38], explaining the XANES results (Fig. 5c), and the remaining low-activity S2 sites show poor selectivity for CRR.

We further experimentally confirm the Fe demetallation by quantifying the presence of iron in the electrolytes from cells performing isolated OER, RMORs or coupled CRR-RMOR (Supplementary Tables 5–7, and Notes 14–16). Notably, the orange-colored electrolyte after OER in contrast to the colorless electrolyte after HzOR provides direct visual evidence (Supplementary Fig. 29). Note that the scanning electron microscopy (SEM) images of the cathodes after CRR and various battery operations show inherited nanoparticle morphology, implying that the carbon framework of Fe-N-C is well preserved (Supplementary Figs. 30 and 31).

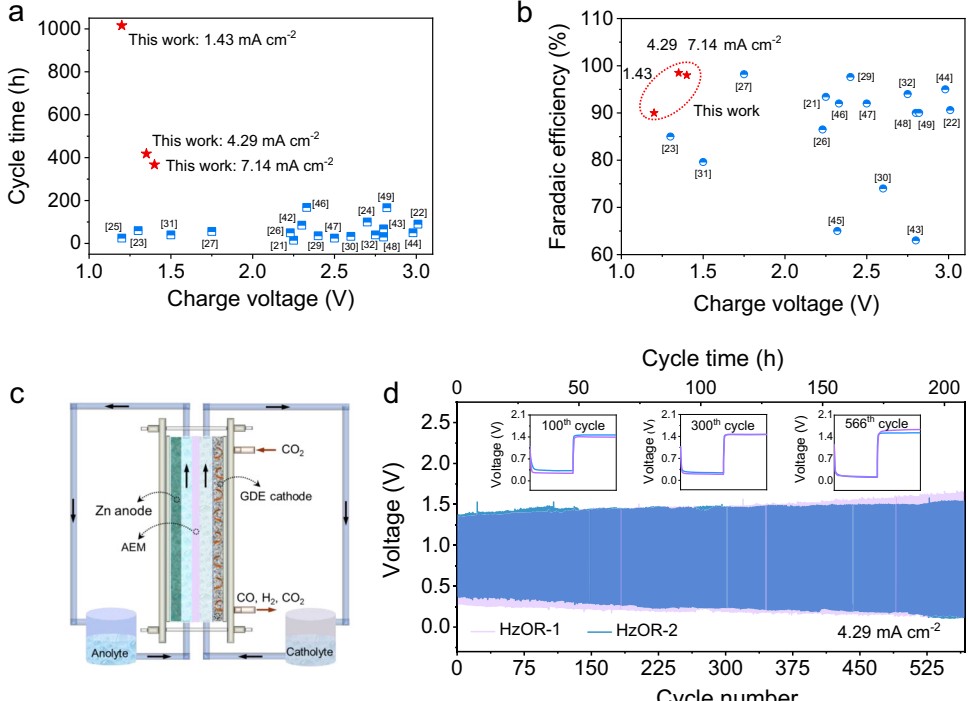

**Fig. 6 | Performance metrics and a proof-of-concept demonstration of the flow cell.** The electrochemical performance of the Zn-CRR/HzOR batteries that greatly outperforms the reported Zn-CO₂ batteries, regarding the charging voltage as a function of cycle time (**a**) and the FE of CO or formic acid generation (**b**). **c** Schematic of the flow cell. **d** The long-term stability of two nearly identical flow cells based on HzOR (HzOR-1 and HzOR-2), indicating high reproducibility in the range of used flow rates: the flow rate of catholyte were 13.4 and 26.8 rpm for HzOR-1 and HzOR-2, respectively, and the anolyte flow rate remained at 20 rpm.

One dangling question is why the Zn-CRR/HzOR battery still shows high performance despite a slight degree of Fe demetallation (Supplementary Tables 5–7). We here propose two assumptions: a) the Fe demetallation may occur on the S2 sites owing to the electron-donating feature of Hz molecules and the low charging voltage, and thus the S1 sites remain majorly active; and b) even if Fe demetallizes from S1 sites, electrochemical transformation of the S2 sites to S1 sites may occur. Both assumptions are possible as indicated by a shorter Fe-N distance (1.42 Å) and less oxidized state of Fe. Further study into these assumptions is required in the future. Taken together, highly active Fe sites in Fe-N-C account for the long-term operation of the Zn-CRR/HzOR battery.

## Scalability demonstration

There has been a lack of demonstration for scalability in the field of Zn-CO₂ batteries. Given that our Zn-CRR/HzOR battery dramatically outperforms all reported Zn-CO₂ batteries in terms of the cycling life, charging voltage, and FE of the aimed product (Fig. 6a, b, and Supplementary Table 2)[21–27,29–32,42–49], we now demonstrate the scalability of our battery that couples CRR and HzOR.

For such, a scaled flow cell configured with a gas diffusion electrode (GDE) and an anion exchange membrane (AEM) was assembled, as illustrated in Fig. 6c. The dynamic equilibrium between CO₂ gas and catholyte is significant for the durability of GDE. The cells using optimized catholyte flow rates (13.4 and 26.8 rpm) are denoted as HzOR-1, and HzOR-2. Both cells showed an excellent rechargeability with a small discharging–recharging voltage polarization of ~1.0 V over a continuous operation for 566 cycles (207.7 h, Fig. 6d). The unprecedented performance shown here signifies the practicability and scalability of our Zn-CRR/HzOR battery.

## Selection criteria of reducing molecules for future design

A prerequisite for the molecules is the strongly reducing nature, which is evidenced by the rapid fading of the investigated Zn-CRR batteries

that are based on the oxidation of non-reducing methanol, ammonia, and urea (Supplementary Fig. 32 and Note 17). In addition to the reducing molecules of Hz, ACTO, and TUDO discussed above, another two reducing molecules, butanone oxime (BTO) and acetaldoxime (AAO) show enhanced cycling performance in the Zn-CRR/RMOR batteries, illustrating the generality of reducing molecules effects (Supplementary Fig. 33).

The second consideration is that the oxidation products of reducing molecules should preferentially be easily separated and collected from the system in real-time. Our Zn-CRR/HzOR battery shows an outstanding cycling life without the need for periodic solution purification owing to the gaseous product N₂ that endows integrity of the system (detailed discussion in Supplementary Figs. 13, 14, and Note 5). A less favorable case is the formation of carbonaceous organics such as acetone and urea, which can poison the active sites and cause the battery failure should no periodic regeneration be installed (details in Supplementary Figs. 13, 15, 34, and Notes 5, 6, 18).

The other requirement is that the oxidation potential of reducing molecules needs to be as low as possible for high energy efficiency, which is critical for this new technology to achieve synergistic CO₂ utilization and energy storage in an energy-economical way.

## Discussion

By coupling CRR with the oxidation of reducing molecules instead of OER, we demonstrated the long cycling life and high energy efficiency for the Zn-CRR/RMOR batteries. We showed that this approach is universal for various reducing molecules, and revealed that the suppressed catalyst degradation by RMOR without oxygenated species at a low charging voltage is the origin of the reducing molecules effect. Remarkable enhancements in the cycling life (2768 cycles) and dramatic decrease in the charging voltage (1.35 V) were achieved with FE_CO as high as 96% for the Zn-CRR/HzOR battery, which is far beyond any reported Zn-CO₂ batteries. Such a strategy can be potentially extended to CO₂ reduction to other high-value products as demonstrated with

the formation of formic acid. Our work provides new insights for a new category of aqueous batteries that are immune to the catalyst structural damage problem caused by OER, and paves an economical way for fulfilling sustainable $CO_2$ utilization, wastewater treatment, and energy storage in one aqueous battery.

## Methods

### Synthesis of the Fe-N-C Catalyst

The Fe-N-C catalysts was prepared by pyrolysis of a Fe-ZIF-8 precursor. In detail, a 150 mL methanol solution of 2-methylimidazole (0.985 g) was mixed with the other 150 mL methanol solution containing 0.8475 g $Zn(NO_3)_2·6H_2O$ and 0.3 g $Fe(NO_3)_3·9H_2O$ as well as 0.02 g home-made carbon sheet. The mixture was first stirred at ambient temperature followed by stirring at 60 °C for obtaining the Fe-ZIF-8 precursor, which was then collected by filtration, washed with ethanol, and dried at 60 °C in a vacuum oven. After carbonizing the Fe-ZIF precursor in a tube furnace under steady Ar gas flow at 1100 °C for 1 h, the Fe-N-C catalyst was obtained.

### Characterizations

The morphology, microstructure, and elemental distribution of the Fe-N-C catalyst were investigated by field emission scanning electron microscopy (SEM, Carl Zeiss Microscopy GmbH), transmission electron microscopy (TEM, Thermo Fisher Talos F200s) and aberration-corrected high-angle annular dark-field scanning transmission electron microscopy (HAADF-STEM, FEI Titan Cube Themis G2). X-ray diffraction (XRD) pattern was collected by the X-ray diffractometer with Cu-Kα radiation (Bruker D8 Advance XRD). X-ray photoelectron spectroscopy (XPS) spectra were recorded by Thermo Scientific K-Alpha with monochromatic Al-Kα X-ray, and the C 1s peak set at 284.8 eV was used as internal standard. The contents of C and N in Fe-N-C were measured by element analysis on Elementar (Vario EL cube), and the Fe content in Fe-N-C was quantified by inductively coupled plasma optical emission spectrometer (ICP-OES, Aglient 5110). The dissolved amount of Fe in the electrolytes was determined by inductively coupled plasma mass spectrometry (ICP-MS, Aglient 7800). Raman spectra were recorded on a microscopic confocal Raman spectrometer (WiTech alpha 300R) with an excitation of 532 nm laser light. The X-ray absorption spectra (XAS) including X-ray absorption near-edge structure (XANES) and extended X-ray absorption fine structure (EXAFS) at Fe K-edge were collected at the 4B9B beamline of Beijing Synchrotron Radiation Facility (BSRF) under ring conditions of 2.2 GeV and about 80 mA. A Si (111) double-crystal monochromator was used for energy selection, and the data collection was conducted in transmission mode using ionization chamber for references ($Fe_2O_3$ and $Fe_3O_4$), and in fluorescent mode for the Fe-N-C catalyst and post-cycle cathodes at ambient air. Data processing was performed with the Athena and Artemis modules in the IFEFFIT software packages[50]. The XANES spectra were obtained by background-subtracting and normalizing in the Athena module[50]. The $k^2$-weighted EXAFS data were Fourier transformed (FT) to R-space with different coordination shells, and the fitting for the first-shell was performed with Fe-N scattering path using Artemis module[6,50].

### Working electrode fabrication for the electrochemical measurements

Prior to the preparation of working electrode, the catalyst ink consisting of 2 mg catalyst, 400 μL ethanol, and 10 μL 5% Nafion solution was sonicated for 5 h. After depositing 140 μL of catalyst ink onto the carbon paper (0.7 cm$^2$) and drying at ambient temperature, the working electrode with a mass loading of 1 mg cm$^{-2}$ was prepared.

### H-type electrochemical cell for $CO_2$ reduction

The electrochemical experiments were conducted in a three-electrode configuration. For the linear sweep voltammetry (LSV) measurement and stability test, Nafion N117 membrane was placed in the configuration of H-type cell. During the electrolysis, Pt plate (1 cm × 1.5 cm) and Ag/AgCl (filling with saturated KCl solution) were used as the counter and reference electrodes, respectively. The electrolyte (0.5 M $KHCO_3$) was saturated with $CO_2$ for 20 min at a flow rate of 20 ml min$^{-1}$ before initiating experiments. Electrochemical electrolysis was carried out by electrochemical workstation (Gamry 5000E) and the current density ($J$) was normalized based on the working electrode area (0.7 cm$^2$). LSV tests were collected at 5 mV s$^{-1}$. The measured potentials after $IR$ compensation were referred to the reversible hydrogen electrode (RHE, $E_{RHE} = E_{Ag/AgCl} + 0.197 + 0.0591 × pH$). The pH values of 0.5 M $CO_2$-saturated and Ar-saturated $KHCO_3$ electrolytes were about 7.5 and 9.0, respectively.

### Rechargeable Zn-CRR/RMOR battery

The rechargeable Zn-CRR/RMOR battery was assembled with a gas-tight H-type cell with a double-electrolyte configuration separated by a bipolar membrane (BPM), and the cell cycling experiments were carried out at room temperature. A solution consisting of 1 M KOH and 0.02 M $Zn(CH_3COO)_2$ was used as the anolyte. The 0.5 M $KHCO_3$ catholyte was bubbled with $CO_2$ at a rate of 20 ml min$^{-1}$ controlled by a mass flowmeter. The anode is a zinc plate (3 ×5 cm, thickness: 3 mm). The preparation of cathode was the same to the above working electrodes. The BPM is commercially available and the membrane in this study was purchased from Fuel Cell Store (Fumasep FBM-PKTM). The BPM arrived as a 20 × 30 cm size sheet in wet form and is ready to use. Further details on information of the membrane are described below, some of which is provided by the supplier. The membrane should be stored in 1 M NaCl-solution and avoid to dry out, as micro-cracks may likely occur if shrinking upon drying. A 100 ppm of $NaN_3$ should be added to prevent biological growth if stored for a long period of time. At 100 mA cm$^{-2}$ in 0.5 M NaCl at 25 °C, the membrane shows high water splitting efficiency (>98%) and low water splitting voltage (<1.2 V). Besides, it also exhibits excellent mechanical properties at low thickness (0.13-0.16 mm) and in a wide stability range (pH: 1-14) at 25 °C. In addition, it can also withstand high alkaline corrosion. BPMs consist of a polymeric cation-exchange layer (CEL) (with fixed anions and mobile cations) and an anion-exchange layer (AEL) with fixed cations and mobile anions. During discharging, water dissociation occurs at the AEL/CEL interface, $H_2O → H^+ + OH$, with the $H^+/OH^-$ driven through the CEL/AEL. During charging, water generation reaction occurs at the intermediate layer with $H^+/OH^-$ ions diffusing into the CEL/AEL. The stability study of the BPM was provided in Supplementary Figs. 35, 36, and Note 19. The deteriorated zinc anodes were replaced with new Zn plates (i.e. refreshing the zinc anode) for some batteries as mentioned in the manuscript. One reason for needing refreshing the zinc anode is that the intrinsically inhomogeneous Zn nucleation and formation of dead zinc, while another possible reason is being exposed to the open air may further lead to zinc corrosion in the anolyte. From the perspective of techno-economic analyses, zinc is cheap and readily available compared with the catalysts, bipolar membranes, gas diffusion layers etc. (i.e. low replacement cost).

For the flow cell, the air cathode (0.5 × 2 cm) was prepared using a gas diffusion layer (GDL) with 1 mg catalysts coated on the liquid-facing side, and the anion-exchange membrane (AEM) was used. Correspondingly, the anode is a zinc plate (0.5 × 2 cm, thickness: 0.3 mm). The optimized flow rates of catholyte, anolyte, and gaseous $CO_2$ was controlled at 26.8 rpm, 20 rpm, and 5 ml min$^{-1}$, respectively. Considering that the stability of gas diffusion electrode (GDE) is affected by the catholyte flow rate, we also conducted experiment with catholyte at 13.4 rpm for a comparison. The galvanostatic charge and discharge tests were conducted on a NEWARE battery tester. With the flow cell which is configured with the anolyte fully sealed, there was no need for refreshing the Zn anode during the 200-hour cycling.

## Product analysis

The gas-phase products were quantified by using an online gas chromatograph (GC, FULI GC9790II) equipped with a thermal conductivity detector (TCD) for $H_2$ quantification and two flame ionization detectors (FIDs), in which one is for CO and $CH_4$, and the other is for $C_2$ and $C_3$ products. Ultrapure Ar (99.999%) was used as the carrier gas. The flow rate of $CO_2$ was controlled by a mass flowmeter with a rate of 20 mL min$^{-1}$ at the inlet of electrochemical cell. The absence of liquid products in the electrolyte after electrolysis was confirmed by $^1$H nuclear magnetic resonance (NMR, Bruker, 600 MHz Ultrashield), further confirming CO as the major product. For the NMR measurements, dimethylsulfoxide (DMSO) was used as the internal standard, and 2 ml of the catholyte was mixed with 0.2 mL of $D_2O$ and 1 μL of DMSO for measurement. Faradaic efficiency (FE) was calculated as follows:

$$FE = \frac{J}{J_{total}} = \frac{v \cdot n \cdot F}{J_{total}} \tag{1}$$

where FE: faradaic efficiency for CO or $H_2$;

$J$: partial current density for CO or $H_2$ generation (A cm$^{-2}$);

$J_{total}$: the total current density (A cm$^{-2}$), which was quantified by Gamry 5000E;

$v$: the production rate of CO or $H_2$ (mol s$^{-1}$), which was quantified by GC;

$n$: the transferred number of electrons, which is 2 for CO and $H_2$;

$F$: Faraday constant, 96485 C mol$^{-1}$.

## Computational methods

Density functional theory (DFT) calculations were performed to disclose the mechanism of the high selectivity of CRR. All the structures were optimized with the Vienna ab initio simulation package[51,52] (VASP. 5.4.4). The Perdew–Burke–Ernzerhof[53] (PBE) functional and the projector augmented wave (PAW) potentials[54] were employed in all the density-functional theory (DFT) calculations. The Van der Waals effects were considered using Grimme's DFT-D3[55] correction. The cut-off energy was set as 500 eV for plane wave expansion and the Brillouin zone sampling was conducted using $2 \times 2 \times 1$ Γ-centered k-point mesh. The convergence criteria for energy and force in the geometry optimizations were 10$^{-6}$ eV and 0.01 eV Å$^{-1}$, respectively. A vacuum space of 15 Å was applied to ensure negligible interaction along z-direction. All the calculations were spin-polarized. The Gibbs free energy change of each step (ΔG) for CRR was calculated under the theoretical framework of standard hydrogen electrode (SHE) developed by Nørskov and co-workers[56].

## Data availability

The data generated in this study are provided in the Supplementary Information/Source Data file. Source data are provided with this paper.

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

## Acknowledgements

This work was financially supported by the National Key Research and Development Program of China (Grant No. 2022YFE0198600, Q.P.), National Natural Science Foundation of China (Grant No. 22209003, Y.L.; 22075002, Q.P.), the China Postdoctoral Science Foundation Funded Project (Grant No. 2021TQ0007, Y.L.; 2021M700212, Y.L.), and the Beijing Natural Science Foundation (No. Z220020, Q.P.). All NMR experiments were performed at the Beijing NMR Center and the NMR facility of National Center for Protein Sciences at Peking University. We appreciate X.N. (Beijing NMR Center, Peking University) for the NMR test and analysis.

## Author contributions

Y.L. and Q.P. conceived the concept. Y.L. designed catalyst synthesis and performed the electrochemical work, with assistance from L.Z. Y.A. designed and performed DFT calculations as well as data analysis. Y.L. characterized the catalyst and analyzed the data. X.L. and P.G. performed the atomic-resolution HAADF-STEM measurements. Y.L., J.Z., and Q.P. together performed the XAS measurements, data analysis and fitting, and interpretation of the XANES and EXAFS results. All authors have thoroughly discussed the data. Y.L., Q.P., G.H. wrote and edited the manuscript with feedback from all the contributing authors. Q.P. acted as the project supervisor.

## Competing interests

The authors declare no competing interests.
