## [Peer Review File · Nature Communications]

Integrated energy storage and CO₂ conversion using a long-cycle-life aqueous battery with tamed asymmetric reactionsEditorial Note: This manuscript has been previously reviewed at another journal that is not operating a transparent peer review scheme. This document only contains reviewer comments and rebuttal letters for versions considered at *Nature Communications*.

REVIEWER COMMENTS

Reviewer #1 (Remarks to the Author):

Generally, all of the comments have been addressed. Now, it is suitable for publication in *Nature Communications*.

Reviewer #4 (Remarks to the Author):

This work actually reports a rechargeable Zn-based battery with catalytic CO₂ reduction and hydrazine oxidation at the cathode for discharging and charging process, respectively, to realize a low charging voltage of the battery by taking advantage of low oxidation potential of Hydrazine oxidation. I mentioned that similar work has been reported previously (*Adv. Mater.* 2022, 34, 2207747), which shows charging voltage lower than 0.6 V at 5 mA cm⁻² with catalytic H₂O reduction and hydrazine oxidation at the cathode for discharging and charging process, respectively. However, it should be noted that complete investigations have been carried out in this work including the excellent battery performance and mechanistic insights on enhanced CRR activity with hydrazine addition at the cathode. Therefore, I can recommend its publication after addressing following questions.

1. In Fig. 3f, it seems that the CO FE increases with the cycle number no larger than 500, it is strange and should provide an explanation.
2. When calculation the adsorption energies of *CO and the free energy for each step involved in CRR, it is found that Fe-N-C absorbed with an additive molecule is built as model for calculation. First, it should to investigate the adsorption behavior of these additive on the surface of Fe-N-C. Second, in my opinion, this kind of calculation is not very accurate because the additive molecules are dispersed in the electrolyte solution instead of adsorbed on the catalyst surface. Finally, the effect of number or concentration of additive molecules on reaction energies for CRR should be explored as well.
3. Following above question, the decreased CRR performance arisen from ACTO and TUDO should be further explored, which is important to exclude the inefficient additives. From Fig. 4, the rate-determining step should be the desorption of *CO, however, it is the same for ACTO and no ACTO addition. Besides, the competitive reaction, hydrogen evolution is suppressed by ACTO.
4. For Supplementary Table 2, it is suggested to provide the electrolyte types for reported Zn-CO₂ batteries for better comparison as the electrolyte can affect the battery performance.
5. Compared to the initial one, the cycled Fe-K-edge shifts to the smaller energy in Fig. 5c, indicating that valence of Fe becomes smaller though this energy is slightly larger than that for Hz additive. Does it indicate that Hz can reduce partial FeO_x to FeN_x? If so, whether Fe-N-C can be regenerated?
6. For Fig. 1b, it is suggested to show the difference of "stable" and "ultra stable".
7. For Fir. 1b, the atomic structure for second molecule at left is different from the right one.
8. For Supplementary Figure 22, why a hydrogen atom disappeared after absorbed on Fe-N-C while other molecules are not?
9. How about the CO FE with the implementation of RMORs at the cathode?

10. With the use of anion exchange membrane, OH⁻ may transfer to the cathode, will it affect the CRR performance of the Fe-N-C?

Point-by-Point Response to Reviewers' Comments

We deeply appreciate the reviewers' insightful and constructive comments. According to your suggestions on our previous manuscript, we have carefully taken all the comments into consideration in preparing our revision and corrected the technical issues. All the queries have been answered with a point-by-point response to the concerns below (in blue). Our revisions are also highlighted with yellow in the revised manuscript.

Reviewer #1 (Remarks to the Author):

Generally, all of the comments have been addressed. Now, it is suitable for publication in Nature Communications.

Our response:

We truly appreciate the reviewer's insightful comment on our work.

Reviewer #4 (Remarks to the Author):

This work actually reports a rechargeable Zn-based battery with catalytic CO₂ reduction and hydrazine oxidation at the cathode for discharging and charging process, respectively, to realize a low charging voltage of the battery by taking advantage of low oxidation potential of Hydrazine oxidation. I mentioned that similar work has been reported previously (*Adv. Mater.* 2022, 34, 2207747), which shows charging voltage lower than 0.6 V at 5 mA cm⁻² with catalytic H₂O reduction and hydrazine oxidation at the cathode for discharging and charging process, respectively. However, it should be noted that complete investigations have been carried out in this work including the excellent battery performance and mechanistic insights on enhanced CRR activity with hydrazine addition at the cathode. Therefore, I can recommend its publication after addressing following questions.

We truly thank the reviewer for the evaluation. The constructive comments now have allowed us to further improve the quality of our manuscript, as detailed below.

As a brief response to the reviewer's comment on the similarity with the previous report (*Adv. Mater.*, 2022, 34, 2207747), we would like to clarify that the chemistry proposed in our work is in fact very different. The key concept in the refereed work is the discovery of a high-performance bifunctional catalyst of 3D Mo₂C/Ni@C/CS that allows low charging voltage < 0.6 V at 5 mA cm⁻² and long-term stability for 600 cycles (200 h) with H₂ generation (discharge: 2H₂O + 2e⁻ → H₂ + 2OH⁻) and hydrazine oxidation (charge: 1/2 N₂H₄ + 2OH⁻ → 1/2 N₂ + 2H₂O + 2e⁻). The reactions involved H₂ generation and hydrazine oxidation.

However, the concept of reaction design and the function of the device in our work is different from the refereed article in terms of scientific challenge and mechanism.

Our work explored the reduction of CO₂ in a rechargeable Zn battery. Although hydrazine is similarly used in our work, the challenge is different. The major challenge in successfully integrating energy storage and CO₂ conversion using aqueous zinc batteries

lies on the catholyte chemistry on recharging and its consequences on the catalyst. As we revealed in our work, the high charging voltage of OER leads to irreversible Fe-demetalation of the Fe-N-C catalyst, and replacing the OER with the designed low-voltage HzOR significantly reduces this process.

In detail, altering of CRR and OER upon dis(charge) periodically disturbs the redox environment around the catalyst, and the irreversible degradation would accumulate over each cycle and eventually leads to permanent structural degradation. This has not been revealed in the previously reported Zn-CO₂ batteries. With thorough understanding of the degradation mechanism, we solved this challenge by replacing OER with HzOR for a long cycling life, low charging voltage, and high FE_{CO}.

For the refereed work, two aspects should be noted: a) at small overpotentials (< 0.4 V vs. RHE), H₂ generation is more energetically favorable than CRR for most catalysts (*Nat Commun.*, **2023**, 14, 6164; *J. Am. Chem. Soc.*, **2021**, 143, 19417–19424; *Angew. Chem. Int. Ed.*, **2021**, 60, 7607–7611; *Adv. Energy Mater.*, **2020**, 10, 2002499); and b) the alkaline solution (1 M KOH as catholytes) benefits the process of N₂H₄ oxidation (*Adv. Mater.*, **2022**, 34, 2204388; *Angew. Chem. Int. Ed.*, **2021**, 60, 5984–5993). Therefore, these two aspects inherently lower the charging voltage, in contrast to the 0.5 M KHCO₃ electrolyte that we had to employ for CRR in our work. In our Zn-CRR/RMOR battery, the adopted catalyst needs to not only switch periodically between CRR reduction (discharge) and oxidation (charge) in 0.5 M KHCO₃, but also maintain high selectivity of CO, which is technically more challenging.

Therefore, from the conceptual design to device functions, our work featuring the integration of energy storage and CO₂ conversion is a different scenario and poses different technical challenges with the refereed work. With no intention to compare the two works, we hope this can explain the major challenge we are aiming to resolve in the field.

Comment #1: In Fig. 3f, it seems that the CO FE increases with the cycle number no larger than 500, it is strange and should provide an explanation.

Our response:

We appreciate the reviewer's insightful comment on the increase in the FE_{CO} over cycling.

The increase in FE_{CO} is related to the slow activation process as the employed current density is small (1.43 mA cm⁻²). In fact, the increase in FE_{CO} from the initial value of ~ 90% to ~ 94% (at about 500 cycles) is not considered significant. We frequently observed this phenomenon in our lab when measuring CRR at a small current density, and thus we believe there is an activation process here (as has been also discussed in some previous reports, e.g. *Nat Commun.*, **2021**, 12, 3765; *Angew. Chem. Int. Ed.*, **2021**, 60, 12554–12559; *Nat Catal.*, **2023**, 6, 939–948).

Comment #2: When calculation the adsorption energies of *CO and the free energy for each step involved in CRR, it is found that Fe-N-C absorbed with an additive molecule is built as model for calculation. First, it should to investigate the adsorption behavior of these additive on the surface of Fe-N-C. Second, in my opinion, this kind of calculation is not very accurate because the additive molecules are dispersed in the electrolyte solution instead of adsorbed on the catalyst surface. Finally, the effect of number or concentration of additive molecules on reaction energies for CRR should be explored as well.

Our response:

We appreciate the reviewer's constructive comment.

→To answer the first question: we have now calculated the adsorption energy of these additives (Hz and ACTO) on the surface of Fe-N-C. The result in Figure R1 shows that the adsorption energy for ACTO and Hz are -0.72 and -0.74 eV, respectively. Both are negative, indicating the adsorption behavior is a spontaneous process.

Figure R1. The calculated adsorption energy of ACTO and Hz on the Fe-N-C catalyst.

→To answer the second question: we agree that a large proportion of additive molecules are dispersed in the electrolyte solution as there is limited number of sites for adsorption. However, based on the above result of negative adsorption energy (-0.72 and -0.74 eV), we believe the additive molecules can adsorb on to the catalyst surface, and benefits the CO₂-to-CO reduction. This is in alignment with both of DFT calculation and experimental results.

Such model has been widely used by other works (*Nature Chem.*, **2009**, 1, 37–46; *Adv. Catal.*, **2000**, 45, 71-129; *Nat Commun.*, **2022**, 13, 3158). For example, a previous report described a metallic copper catalyst that showed high selectivity for CO₂-to-CH₄ conversion with controlled surface reconstruction by an electrolyte additive (ethylenediamine tetramethylenephosphonic acid, EDTMPA) (*Nat Commun.*, **2022**, 13, 3158). The report was based on the adsorbed model which helped unveil very important information on the occupation of active sites, indicating such model is reliable and reproducible to reveal the catalysis mechanism in studies on catalysts.

→To answer the third question: this is a great question! We have in fact *already* validated the appropriate concentration range for each reducing molecule, as shown in Fig. 2d, e and Supplementary Figure 8. As exemplified by the HzOR, the results show

that high FE_{CO} in a wide range of concentration (23 – 344 mM) is obtained without sacrificing the discharge voltage, indicating little changes in the reaction energies for CRR. However, when the Hz concentration at 459 mM (Supplementary Figure 8a), FE_{CO} decreased by ~ 80% and FE_{H_2} increased by ~ 16%, which should be due to overly high occupation of the active sites by Hz and a change in pH value of electrolytes. Besides, similar trend is also observed in the case of Hz concentration at 688 mM (Supplementary Figure 8b).

We have now added these new experiments and discussions in the revised manuscript.

Comment #3: Following above question, the decreased CRR performance arisen from ACTO and TUDO should be further explored, which is important to exclude the inefficient additives. From Fig. 4, the rate-determining step should be the desorption of $*CO$, however, it is the same for ACTO and no ACTO addition. Besides, the competitive reaction, hydrogen evolution is suppressed by ACTO.

Our response:

We appreciate the reviewer's insightful comments. In fact, the free energy of $*CO$ adsorption for ACTO and Hz addition is the same, which is lower than that without additives. Therefore, the CRR kinetics of ACTO is considered similar with that of Hz. For hydrogen evolution reaction (HER), it is also suppressed with the presence of ACTO. However, an amount of HCOOH product was also found for ACTO by NMR measurement (Supplementary Figure 13b). Hence, ACTO additive may lead to improved CRR kinetics, suppressed HER, but slightly lower FE_{CO} . Therefore, we did not intend to exclude ACTO as an effective additive in the manuscript.

As for TUDO, it first decomposes into urea and H_2SO_3 , and then H_2SO_3 provides H^+ (details in Supplementary Figures 13-14). The H^+ released from TUDO leads to a lower FE_{CO} compared to that on HzOR- FeN_4 and ACTOR- FeN_4 sites, which has been confirmed by the experiments (Supplementary Figure 23 and Note 12). It clearly illustrates that the CRR activity was dramatically suppressed with the additional H^+ , and further the FE_{CO} gradually increased and fully recovered with gradual consumption of H^+ . These experiments confirm that the extra H^+ derived from TUDO is the cause of the decrease in FE_{CO} .

We have now emphasized on these discussions in the revised manuscript.

Comment #4: For Supplementary Table 2, it is suggested to provide the electrolyte types for reported Zn- CO_2 batteries for better comparison as the electrolyte can affect the battery performance.

Our response:

We appreciate the reviewer's insightful comment. Now, we have added the details of electrolyte including catholyte and anolyte in the revised Supplementary Table 2.

Comment #5: Compared to the initial one, the cycled Fe-K-edge shifts to the smaller energy in Fig. 5c, indicating that valence of Fe becomes smaller though this energy is slightly larger than that for Hz additive. Does it indicate that Hz can reduce partial FeO_x to FeN_x ? If so, whether Fe-N-C can be regenerated?

Our response:

We appreciate the reviewer's valuable comment. We fully agree with the reviewer that the shift of Fe-K-edge to a lower energy after cycling indicates there is slight reduction of Fe during the cycles due to the subtle reconstruction in local coordination environments, as we discussed in Supplementary Note 13. This has been observed in a few previous studies (*Nat. Catal.*, **2020**, 4, 10-19; *Science*, **2019**, 364, 1091-1094; *Nat. Catal.*, **2022**, 5, 854-866). However, we note that the Fe-K-edge spectra for HzOR-based cathodes at charged states almost overlap with those at discharged states, suggesting excellent catalyst stability that is immune to both CRR reduction and Hz oxidation, as shown in Fig. 5d. This answers the question that the slight reduction of Fe-N-C over the cycles can be well regenerated and function over hundreds of cycles. This is also evidenced by the almost identical voltage profiles (first four cycles) for the Zn-CRR/HzOR batteries (Fig. 5a).

In fact, FeO_x is involved only in the case of OER (additive free), and cannot be regenerated. Specifically, the reactive oxygen species OH^* produced during OER first adsorbs on FeN_4 sites, and the OH^* weakens Fe-N bonding and further triggers the leaching of Fe in a way of FeO_x (*Nat. Energy.*, **2022**, 7, 652-663; *Angew. Chem. Int. Ed.*, **2021**, 61, 202111683; *PNAS.*, **2021**, 118, 2110036118; *Adv. Mater.*, **2022**, 34, 2107421) We do not think there is FeO_x involved in the case of HzOR during the cycling process, in which Fe-N active sites are well maintained in two possible routes as we discussed in the original manuscript (please see page 16 and lines 343-351, and lines 359-367).

Comment #6: For Fig. 1b, it is suggested to show the difference of “stable” and “ultra stable”.

Our response:

We appreciate the reviewer's valuable comment.

We propose to define in this way: in contrast to the case of OER with <10 cycles (i.e. unstable) in (b), the oxidation of ACTO refers to an enhanced cycling performance of >100 cycles (i.e. stable), while HzOR is a more favorable reaction with a long-term cycling stability of >1000 cycles (i.e. ultra-stable).

Now, we have described the difference between “stable” and “ultra stable” in the revised manuscript (please see the caption of Fig. 1b, page 5 and lines 104-107).

Comment #7: For Fir. 1b, the atomic structure for second molecule at left is different from the right one.

Our response:

We appreciate the reviewer's comment. We think the reviewer might refer to the fact that the oxidation product of TUDO is different from TUDO itself (Figure R2). The schematic is meant to show the main oxidation products of the molecules. For the oxidation of TUDO, SO_3^{2-} is the key product (details are discussed and presented in Supplementary Figures 13-14). Now, we have added annotation in the caption of Fig. 1b for a clear description as follows (please also see page 5 and line 107-108).

Figure R2. Schematics showing the concerted effect of lower charging voltages and promoted catalyst stability with the implementation of RMORs, as well as the underlying mechanism from the view of energy landscape. It is noted that the schematic is meant to show the main oxidation products of the molecules. The red, white, grey, blue, orange, and yellow balls represent O, H, C, N, Fe, and S atoms, respectively.

Comment #8: For Supplementary Figure 22, why a hydrogen atom disappeared after absorbed on Fe-N-C while other molecules are not?

Our response:

We truly appreciate the reviewer for the valuable reminder. The hydrogen atom is indeed present, but as the hydrogen atom is in white, it could be distinguishable but weak/dim in a certain visual direction. Now, we have revised our geometric configurations shown in Figure R3 below, please also see page S23 and Supplementary Figure 22.

Figure R3. Computational results of geometric configurations with reducing molecules adsorbing on the FeN₄ sites of a Fe-N-C catalyst. (a-b) The coordination of Fe active sites with ACTO and Hz molecules adsorbed on FeN₄ sites, which is different from that in blank FeN₄ sites during CRR (a) and HER (b) processes. The ACTO and Hz apparently interact with the Fe active sites and weaken the interaction of *CO with Fe, thus facilitating the desorption of CO (Orange, blue, grey, red, and white balls stand for Fe, N, C, O, and H atoms).

Comment #9: How about the CO FE with the implementation of RMORs at the cathode?

Our response:

Our work has extensively studied and recorded FE_{CO} in the cell with RMORs at the cathode, as shown in Fig. 2, Fig. 3, and Fig. 6, and thoroughly discussed in the manuscript. As a response to the reviewer's comment, herein, we would like to present the main conclusions of FE_{CO} as follows:

By replacing the conventional high-potential and kinetically sluggish OER using hydrazine oxidation (HzOR) on recharging, our aqueous battery demonstrates an unprecedented long operation life over 1000 hours with a charging voltage as low as 1.2 V. Specifically, as shown in Fig. 3, the Zn-CRR-HzOR battery can operate for 1000 cycles at 4.29 mA cm^{-2} ; critically, the FE_{CO} maintained about 97% and 92% at 100 and 1000 cycles respectively. In contrast, the "Zn-CO₂" battery relying on OER could operate for only 8 hours, and even so the FE_{CO} quickly dropped to 80% and retained only 14% after 22 cycles. The lifespan of our Zn-CRR/HzOR battery increases by up to 50-fold. Further, the cell using HzOR shows a very low recharging voltage of 1.35 V, much lower than that using OER (2.6 V). This greatly improves the round-trip energy efficiency (EE) for the battery to function as energy storage. Further, the battery shows an outstanding cycling life of 2768 cycles (1015.8 h) at a moderate current density of 1.43 mA cm^{-2} .

We further place our work in the frame of literature for comparison with the state-of-the-art research in the field. As seen from the Fig. 6a, b and Supplementary Table 2, in terms of charging voltage/cycle time, our work demonstrated obvious superiority. Most

of the literature reports on “Zn-CO₂” battery showed an overly high charging voltage of 2.0 ~ 3.5 V with less than 200 hours of cycling, and the four reports showed low charging voltage of 1.2 ~ 1.8 V but with even shorter cycling time of less than 100 hours. In great contrast, our Zn-CRR/HzOR battery can work with low charging voltage of less than 1.4 V for as long as over 1000 hours.

Comment #10: With the use of anion exchange membrane, OH⁻ may transfer to the cathode, will it affect the CRR performance of the Fe-N-C?

Our response:

We truly appreciate the reviewer’s comment. With the anion exchange membrane (AEM), the impact of CRR performance of the Fe-N-C in cathode side can be neglected because the transferred OH⁻, if any, will be rapidly consumed by HCO₃⁻ and CO₂.

Nevertheless, the HCO₃⁻ in the catholyte (0.5 M KHCO₃) transferring to the anolyte (1 M KOH + 0.02 M Zn(CH₃COO)₂) can cause the change in pH value of anolyte. This may hinder the stripping/plating of Zn anode as Zn dissolution prefers a basic solution. Over cycling, the degradation of Zn anode will lead to a significant reduction in both the discharged voltage and cycling life. The catholyte/anolyte stability challenge of crossover could be greatly alleviated by applying a bipolar membrane (BPM) that separately presents proton and hydroxyl to the two chambers, as discussed in the part of Method (please see page 21 and lines 484-489).

Now, we have added this discussion in the revised manuscript (page S35 and lines 555-562).

REVIEWERS' COMMENTS

Reviewer #4 (Remarks to the Author):

The authors have successfully addressed my concerns and the paper is now in a good format. I think the paper can be accepted as it is.

Point-by-Point Response to Reviewers' Comments

Reviewer #4 (Remarks to the Author):

The authors have successfully addressed my concerns and the paper is now in a good format. I think the paper can be accepted as it is.

Our response:

We truly appreciate the reviewer's insightful comment on our work.